# Quality of Monitoring Optimization in Underwater Sensor Networks through a Multiagent Diversity-Based Gradient Approach

**DOI:** 10.3390/s23083877

**Published:** 2023-04-11

**Authors:** Mohamed Ould-Elhassen Aoueileyine, Hajar Bennouri, Amine Berqia, Pedro G. Lind, Hårek Haugerud, Ondrej Krejcar, Ridha Bouallegue, Anis Yazidi

**Affiliations:** 1Innov’COM Laboratory, Higher School of Communication of Tunis (SUPCOM), University of Carthage, Ariana 2083, Tunisia; 2Smart Systems Laboratory (SSL), ENSIAS, Rabat IT Center, Mohammed V University, Rabat BP 713, Morocco; 3Department of Computer Science, OsloMet—Oslo Metropolitan University, 0176 Oslo, Norway; 4Center for Basic and Applied Research, Faculty of Informatics and Management, University of Hradec Kralove, 500 03 Hradec Kralove, Czech Republic; 5Institute of Technology and Business in Ceske Budejovice, 370 01 Ceske Budejovice, Czech Republic; 6Malaysia Japan International Institute of Technology (MJIIT), University Teknologi Malaysia, Kuala Lumpur 54100, Malaysia

**Keywords:** underwater communications, quality of monitoring, diversity, detrimental point process

## Abstract

Due to the complex underwater environment, conventional measurement and sensing methods used for land are difficult to apply directly in the underwater environment. Especially for seabed topography, it is impossible to perform long-distance and accurate detection by electromagnetic waves. Therefore, various types of acoustic and even optical sensing devices for underwater applications have been used. Equipped with submersibles, these underwater sensors can detect a wide underwater range accurately. In addition, the development of sensor technology will be modified and optimized according to the needs of ocean exploitation. In this paper, we propose a multiagent approach for optimizing the quality of monitoring (QoM) in underwater sensor networks. Our framework aspires to optimize the QoM by resorting to the machine learning concept of *diversity*. We devise a multiagent optimization procedure which is able to both reduce the redundancy among the sensor readings and maximize the diversity in a distributed and adaptive manner. The mobile sensor positions are adjusted iteratively using a gradient type of updates. The overall framework is tested through simulations based on realistic environment conditions. The proposed approach is compared to other placement approaches and is found to achieve a higher QoM with a smaller number of sensors.

## 1. Introduction

Quality of monitoring (QoM) [1,2,3,4] of information in the underwater domain [5,6] is a recent concept that has attracted the attention of researchers in the field of wireless sensors and connected objects. In recent years, there has been an upsurge of interest in underwater wireless sensor networks (UWSNs). UWSNs are made up of several autonomous sensor nodes. These sensor nodes are scattered underwater to carry out detection tasks in order to collect different properties related to underwater environments [6]. UWSNs admit a large set of applications that includes, for instance, monitoring the living conditions of fish, such as measuring temperature, humidity, pH, and CO_2_ concentrations, in order to associate those metrics with the amount of fish produced under these conditions and relative to a given time. There are five types of wireless sensor networks, depending on the environment. Terrestrial WSNs are used for communicating base stations efficiently and comprise thousands of wireless sensor nodes deployed either in an unstructured (ad hoc) or structured (preplanned) manner. In an unstructured mode (ad hoc), the sensor nodes are randomly distributed within the target area that is dropped from a set plane [7]. In WSNs, the battery power is limited; however, the battery is provided with solar cells as a secondary power source. The conservation of energy of the WSNs is achieved by using low-duty cycle operations, optimal routing, minimizing delays, and so on. In terms of deployment, maintenance, equipment cost considerations, and careful planning, underground wireless sensor networks are more expensive than terrestrial WSNs [8]. The underground wireless sensor networks (UWSNs) comprise several sensory nodes that are hidden in the ground to observe underground conditions. Additional sink nodes are located above the bottom to transfer information from the sensor nodes to the base station. These underground WSNs deployed into the ground are difficult to recharge. The sensor battery nodes equipped with limited battery power are also difficult to recharge. Additionally, the underground environment makes wireless communication a challenge because of the high attenuation and signal loss level [9]. More than 70% of the Earth is occupied by water. These networks contain several sensor nodes and vehicles deployed underwater. Autonomous underwater devices and vehicles are used to collect data from these sensor nodes. Challenges of underwater communication may be a long propagation delay and bandwidth and sensor failures. Underwater, WSNs are equipped with a limited battery that cannot be recharged or replaced. The difficulty of energy conservation for underwater WSNs involves the development of underwater communication and networking techniques [10]. Multimedia WSNs are proposed to enable tracking and monitoring of events in the sort of multimedia, such as video, imaging, and audio. These networks contain low-cost sensor nodes equipped with cameras and microphones. These sensory nodes of multimedia WSNs are interconnected together over a wireless connection for data retrieval, data compression, and correlation [11]. The challenges with the multimedia WSNs include high bandwidth requirements, high energy consumption, processing, and compressing techniques. Additionally, multimedia contents need high bandwidth for the content to be delivered properly and easily. Mobile WSNs comprise a group of sensor nodes that can be moved on their own and can interact with the physical environment. The mobile nodes can also compute sense and communicate respectively. Mobile wireless sensor networks are much more versatile than static sensor networks. The benefits of mobile WSNs over static WSNs include better and improved coverage, superior channel capacity, better energy efficiency, and so on. Previous solutions for terrestrial wireless sensor networks (TWSNs) cannot usually be applied in UWSNs, as means for transmission using radio frequency (RF) and optical signals are inhibited in underwater environments. Only acoustic signals are viable means for data transmission in a marine environment [12]. The majority of monitoring solutions in both TWSNs and UWSNs neglect the problem of redundancy and correlation among the different measurements. Particularly within UWSNs, the energy cost of sending the data collected is very high; therefore, it is essential to apply some intelligent computation techniques to minimize redundancy and reduce the amount of unnecessary traffic. This is one of the main research aspects in the field of QoM. Moreover, monitoring of the marine environment has gained increasing attention, mainly due to the growing concerns about climate change [13]. In many cases, it is desirable to deploy moving sensors to collect as much information as possible and to optimize the QoM. The sensors need, in many applications, to submerge for data collection. As opposed to this, the use of fixed sensors in buoys on the surface of the water does not provide enough geographical coverage. The movement of sensors by means of underwater robots has catalyzed significant research in robotics and constitutes an enabler for novel schemes within the field of underwater communication. Figure 1 provides an overview of the AUVs used in UWSNs [14,15,16,17]. AUVs are categorized based on their size, depth capability, onboard horsepower, and whether they are all-electric or electrohydraulic, as follows: Micro-class AUVs are very small in size and weight. They can weigh less than 3 kg and are used as an alternative to a diver, specifically in places where a diver might not physically enter, such as a sewer, pipeline, or small cavity. Mini-class AUVs weigh around 15 kg. They are also used as a diver alternative. General-class AUVs have less than 5 HP (propulsion), manipulators, grippers, and a sonar unit used on light survey applications. The maximum depth is less than 1000 meters. Light-work classes typically have less than 50 HP (propulsion). They carry manipulators and are made from polymers such as polyethylene rather than conventional alloys. Heavy-work classes typically have less than 220 HP (propulsion) with the ability to carry at least two manipulators. They have a working depth of up to 3500 m. These networks contain multiple sensor nodes and vehicles deployed underwater. Autonomous underwater vehicles and devices are used to collect data from these sensor nodes, and different sizes and designs can be used depending on the mission. Challenges of underwater communication can be long propagation delay and bandwidth and sensor failures. UWSNs are equipped with a limited battery that cannot be recharged or replaced. The difficulty of energy conservation for UWSN involves the development of underwater communication and networking techniques [18,19,20,21,22].

Monitoring in the underwater domain consists of evaluating the water quality, which is a task of utmost importance. Water quality plays a vital role in fish farming. Good water quality helps farmers ensure maximum fish growth, guarantees a high-quality product, and minimizes the diseases and deaths rate. All these factors increase fish production and consequently influence national and international economic growth. Water contains many parameters that can judge its quality. In aquaculture, there are intervals of standard values [23]; if a value of a parameter exceeds the limits, the water quality will be influenced. The main parameters of aquaculture water are dissolved oxygen (DO), j (NH3, NH${}_{4}^{+}$), nitrite (NO${}_{2}^{-}$), nitrate, turbidity, pH, and temperature [24]. Many factors can influence water quality such as biology, physics, and human activities; they make it a very complex, nonlinear, and dynamic system. This kind of system cannot be managed with a classical method. An outdated classification model does not help to achieve better results. Therefore, the use of new technologies by introducing artificial intelligence and machine learning can be an effective solution.

In this paper, we resort to a multiobjective function to quantify the quality of monitoring in underwater wireless sensor networks. The multiobjective function incorporates two objectives: minimizing covariance of sensor readings in order to reflect the idea of reducing redundancy, and maximizing the diversity among the sensor readings in order to reflect the idea of choosing positions that unveil novel information not described by the rest of the sensors. In this sense, the QoM problem can be seen as an instance of an optimal sensor coverage problem. The novelties in this paper revolve around the following:The first objective is to maximize the coverage of the sensor network by optimizing the placement of sensors. This involves designing a distributed algorithm that enables the sensors to communicate with each other and cooperate to achieve maximum coverage of the area of interest.The second objective is to reduce the redundancy of the sensor network by not only considering the correlation between sensors readings but also the diversity of the readings using the concept of determinantal point process.

The closest work to ours is due to Weiler et al. [25]. The latter work investigates adjusting the positions of sensors based on the gradient descent approach. The objective function contains only one term that takes the inverse of the overall covariance between sensors and points to sense. The reason for choosing the inverse is to penalize regions that are covered by more than a sensor and thus better take into account the marginalcontribution of each sensor. The objective function is intuitive; however, it does not have a sound physical meaning. In our work, we take into account diversity as a new term in the objective function without inverting the overall sum of covariance. Our framework contributes to filling this gap in AUV solutions. In this study, we use AUV robots because of their ability to move underwater without needing any external intervention. AUVs are underwater robots that are typically used in mining areas, agriculture applications, and so forth [26]. They are one of the most significant tools for the exploration and application of marine resources [27,28]. An AUV is a self-piloting vehicle that performs a task, and usually, it is equipped with an onboard artificial intelligence system with a set of programmed commands, which can be modified remotely by data or information broadcast by the vehicle’s sensors [29]. Our network of AUVs is characterized by its ability to move horizontally by ejecting water, and to move vertically, the AUVs use a buoyancy control system.

The remainder of the paper is organized as follows. In Section 3, the proposed solution is detailed. A series of experiments and some simulation results are presented to show the validity and relevance of the proposed approaches in Section 4. Section 5 gives some results and comparisons with similar works, and Section 6 concludes this article while giving some directions for future work. In our network, we consider that the AUVs move in 2D; horizontally and vertically by implementing a GRN which is one of a widely used set of methods in different fields such as swarm robotics [25,26,27].

## 2. Related Work

The application of advanced information and communication technology, such as the Internet of Things (IoT) and the various machine learning methods to better manage the behavior of autonomous underwater vehicles (AUVs), is becoming a trend for the purpose of better QoM. For example, the processing and visualization of water quality data can be carried out remotely and in real time using these AUVs. An example of this application is presented in [30], where an underwater environment monitoring system based on UWSNs is introduced. This system was conceived to be able to perform a large quantity of uninterrupted collected data. Further work is presented in [31], which introduces advanced wireless protocols developed for the IoT in order to highlight their adaptability for the WSN application used in water quality monitoring. Many spatial coverage algorithms are surveyed in [32], with a very detailed comparison. For instance, the authors in [18] propose a top-down positioning scheme (TPS) for acoustic UWSNs while ensuring the quality of service of the new reference nodes during the determination phase of well-located nodes based on the gradient method. Furthermore, the latter work presents a new method of estimating the 3D Euclidean distance to facilitate nonlocalized nodes to find more reference nodes in order to become localized.

A distributed coverage control scheme is described in [33], where a density function describing frequency random events with mobile sensors operates within a restricted range specified by a probabilistic model. The algorithm used in this work is based on the gradient, which needs local information on each sensor and maximizes the probabilities of detection of common random events. For a coverage control problem, costs of communication are calculated according to two scenarios of data collection: the first takes the network as a network that collects data from a single source, and the second one identifies the network with multisource data.To model the cost of communication, the authors use the same form of energy consumption.

Our work builds on the work by Detweiler et al. [34], where the authors deployed a gradient-based decentralized controller that dynamically adjusts the depth of a submarine sensor network to improve the QoM. In contrast to our work, which also uses the concept of diversity, the latter work only involves optimizing a particular redundancy function based on the correlation between the different sensors. This solution was implemented to solve the problem of monitoring chromophoric dissolved organic matter (CDOM) in the Neponset River, which feeds into Boston Harbor. The study proved that the controller converges to a local minimum. This controller is adapted to a network of submarine sensors capable of adjusting their depths. The results of simulations and experiments verified the functionality and performance of this system and the algorithm presented. The SALMON (Sea Water Quality Monitoring and Management) [35] presented a concept of a guidance system using AUVs to detect and perform automated analysis of several water quality parameters.

In order to model the quality of surveillance, ref. [36] focused on the theoretical study of spatial and temporal correlations due to the various physical phenomena of wireless sensor deployment in nature. Two schemes were proposed to reveal the time and space dependence under centralized and distributed settings to maximize the overall QoM based on sensing scheduling. The same authors proposed another study in [37] using the nondecreasing submodular function to measure the QoM, but this time they took into account the correlation in the detected data in order to define distributed scheduling schemes that are used to determine a high QoM in a ring cycle sensor array.

In [25], the authors proposed RDBF, which is a relative remote routing protocol that takes into consideration energy saving while minimizing delays in transmission. This work was based on the use of an aptitude factor to determine the degree of relevance of a node to participate in transmitting packets. This aptitude test helps reduce needless transfers by the nodes, which helps reduce power consumption and end-to-end delay, in addition to reducing redundancy by controlling transfer time of multiple senders. However, none of the existing studies use the concept of diversity from machine learning to deal with issues of quality of monitoring in the underwater environment.

In recent years, the need for controlling robots based on artificial intelligence and, more particularly, machine learning instead of programming has increased. Several methods have addressed this demand using genetic algorithms, neural networks, and other artificial intelligence (AI) or machine learning methods to control some of the functionality of robots [28]. The majority of mutlirobot systems rely on a default programmed algorithm, something that cannot be applied in a dynamic environment characterized by unpredictable change; therefore, the robot system has to adapt with the environmental changes and take into account the local perception of the robot. The authors of [29] proposed the Hierarchical Gene Regulatory Network (H-GRNe) for Adaptive Multirobot Pattern Formation, which is a two-layer gene regulatory network (GRN) model that adapts the generation and formation of multirobot patterns. In this model, the adaptation part of pattern generation is conducted in the first layer and then these generated patterns will drive the robots in the second layer with a decentralized control mechanism. The authors accompanied their study with simulation in a changing environment that proved the efficiency of H-GRNe to form the desired pattern, and also a strong adaptation to robot failure. The AUVs used in this network apply the cellular adhesion molecules (CAM) combined with GRN controllers proposed by [33]. This model is based on the control of GRN-CAM hydrons, which refers, in our case, to a group of AUVs.

## 3. An Optimization Function for Water Quality which Minimizes Sensor Redundancy and Maximizes Diversity

This section provides the details of the proposed solution, with the additional aim of highlighting the characteristics of the proposed architecture and how it is implemented. We consider *N* AUVs at locations $P_{i}\left(x_{i},y_{i},z_{i}\right)$ with i=1, …,N. We assume that the sensors move in a two-dimensional plane defined by the *x* and *z* axes, with a fixed *y* coordinate, as seen in Figure 2, reducing the three-dimensional positioning to $p_{i}\left(x_{i},z_{i}\right)$.

We will assume that the correlation between pairs of sensors decreases, not necessarily isotropically, with their distance as a Gaussian function. Consequently, we can postulate that the covariance between two sensors *i* and *j* is given by
(1)$$\mathrm{Cov}\left(p_{i},p_{j}\right)=\exp\left(-\frac{{\left(x_{i}-x_{j}\right)}^{2}}{2\sigma_{x}^{2}}-\frac{{\left(z_{i}-z_{j}\right)}^{2}}{2\sigma_{z}^{2}}\right)$$
where $\sigma_{x}$ and $\sigma_{z}$ have the meaning of (spatial) correlation decreasing rates in the *x* and *z* directions, respectively.

### 3.1. Gradient Based on Covariance

Since we want to maximize redundancy among the sensors, we need to minimize the overall pairwise correlation between sensors. In other words, we minimize the following function:(2)$$H\left(p_{1},\ldots ,p_{N}\right)=\sum_{i=1}^{N}\sum_{j=i+1}^{N}\;\mathrm{Cov}\left(p_{i},p_{j}\right)\;.$$

The minimum of $H(p_{1},\ldots ,p_{N})$ fulfills the equations
(3)$$\begin{array}{ccc} \nabla_{x_{i},z_{i}}H\left(p_{1},\ldots ,p_{N}\right) & \equiv & \left(\frac{\partial H}{\partial x_{i}},\frac{\partial H}{\partial z_{i}}\right)=0\;, \end{array}$$
for i=1,…,N, yielding
(4)$$\left(\sum_{j=i+1}^{N}\mathrm{Cov}\left(p_{i},p_{j}\right)\frac{\left(x_{i}-x_{j}\right)}{\sigma_{x}^{2}},\sum_{j=i+1}^{N}\mathrm{Cov}\left(p_{i},p_{j}\right)\frac{\left(z_{i}-z_{j}\right)}{\sigma_{z}^{2}}\right)=0\;.$$

### 3.2. Gradient of Diversity

Minimizing the function $H(p_{1},\ldots ,p_{N})$ alone leads to a solution which indeed minimizes redundancy, but does not guarantee that one covers the maximum amount of information in the system. In other words, one also needs to take into account the diversity covered by the set of sensors. Assuming all the information of the system can be encoded in the linear correlations observed in the systems, the determinant of the covariance matrix *L* between pairs of sensors is a proper measure of such a total amount of information, since it reflects the total variance of the data collected by the set of sensors. The idea of using the determinant as a measure of diversity is found also in the the theory of determinantal point processes. We therefore consider the covariance matrix *L* with elements $L_{ij}=\mathrm{Cov}\left(p_{i},p_{j}\right)$, as defined in Equation (Equation 1), and seek its maximum, which is a solution of
(5)$$\begin{array}{ccc} \nabla_{x_{i},z_{i}}\det\left(L\right) & \equiv & \left(\frac{\partial \det(L)}{\partial x_{i}},\frac{\partial \det(L)}{\partial z_{i}}\right)=0\;, \end{array}$$
which can be written as
(6)$$\begin{array}{ccc} \left(\frac{\det(L)}{\sigma_{x}^{2}}\mathrm{tr}\left(L⊙L^{-T}\frac{dG}{dx_{i}}\right),\frac{\det(L)}{\sigma_{z}^{2}}\mathrm{tr}\left(L⊙L^{-T}\frac{dG}{dz_{i}}\right)\right) & = & 0\;, \end{array}$$
where *G* is a matrix with elements $G_{ij}=G_{ji}=-{\left(x_{i}-x_{j}\right)}^{2}-{\left(z_{i}-z_{j}\right)}^{2}$ and ⊙ denotes the Hadamard product, for the full derivation of Equation (Equation 6).

### 3.3. Weighted Objective Function

We now combine both the redundancy *H* and diversity *L* in the same weighted objective function *F*, defined as
(7)$$F=w\frac{H-H_{\min}}{H_{\max}-H_{\min}}-\left(1-w\right)\frac{\det\left(L\right)-\det{\left(L\right)}_{\min}}{\det{\left(L\right)}_{\max}-\det{\left(L\right)}_{\max}}\;,$$
where we consider the normalization of both function *H* and *L* to have values between 0 and 1, and introduce a parameter *w* which tunes how much the function *H* dominates over the function *L*.

For simplicity, we define
$$\begin{array}{ccc} N_{H} & = & H_{\max}-H_{\min}\\ N_{L} & = & \det{\left(L\right)}_{\max}-\det{\left(L\right)}_{\min}\\ \zeta & = & \frac{1-\omega }{\omega }\frac{N_{H}}{N_{L}} \end{array}$$
reducing the minimization problem
(8)$$\begin{array}{ccc} \nabla_{x_{i},z_{i}}F & \equiv & 0 \end{array}$$
to
(9)$$\begin{array}{ccc} \left(\frac{\partial H}{\partial x_{i}}-\zeta \frac{\partial \det(L)}{\partial x_{i}},\frac{\partial H}{\partial z_{i}}-\zeta \frac{\partial \det(L)}{\partial z_{i}}\right) & = & 0\;. \end{array}$$

Equation (Equation 9) together with Equations (Equation 4) and (Equation 6) close the optimization problem for extracting the set of locations $\left(x_{i},z_{i}\right)$ of the *N* sensors, which optimizes the redundancy and diversity together. Note that the gradient controller in Equation (Equation 9) converges to a critical point of *F*.

At this juncture, we are ready to present our multiagent algorithm for optimizing the above objective function. From Equation (Equation 9), the numerical implementation of the optimization problem can be performed through a simple Newton–Raphson scheme. Namely, let *t* denote a discrete time instant. We shall update the positions of sensor *i* recursively. The position at time t+1 is given by
(10a)$$\begin{array}{ccc} x_{i}\left(t+1\right) & = & x_{i}\left(t\right)-\lambda \frac{\partial F_{\zeta }}{\partial x_{i}}\;, \end{array}$$
(10b)$$\begin{array}{ccc} z_{i}\left(t+1\right) & = & z_{i}\left(t\right)-\lambda \frac{\partial F_{\zeta }}{\partial z_{i}}\;. \end{array}$$
where λ is a learning parameter.

### 3.4. Derivation of the Gradient of Diversity

From Equation (Equation 1), it is easy to obtain
$$\frac{\partial L_{i,j}}{\partial x_{i}}=-\frac{1}{2\sigma_{x}^{2}}\left(x_{i}-x_{j}\right)L_{i,j}$$

Therefore,
$$\frac{\partial L}{\partial x_{i}}=\frac{1}{\sigma_{x}^{2}}L⊙\frac{dG}{dx_{i}}$$

We apply the Jacob formula:$$\begin{array}{cc} \frac{\partial \det(L)}{\partial x_{i}} & =\det\left(L\right)tr\left(L^{-1}\frac{\partial L}{\partial x_{i}}\right)\\ \; & =\det\left(L\right)tr\left(L^{-1}\frac{1}{\sigma_{s}^{2}}L⊙\frac{dG}{dx_{i}}\right)\\ \; & =\frac{\det(L)}{\sigma_{s}^{2}}tr\left(L^{-1}L⊙\frac{dG}{dx_{i}}\right)\\ \; & =\frac{\det(L)}{\sigma_{s}^{2}}tr\left(L^{-1}L⊙\frac{dG}{dx_{i}}\right) \end{array}$$
and we obtain
(11)$$\frac{\partial \det(L)}{\partial x_{i}}=\frac{\det(L)}{\sigma_{s}^{2}}tr\left(L⊙L^{-T}\frac{dG}{dx_{i}}\right)$$

We used that the diagonal entries of $\left(A\circ B\right)C^{T}$ and $\left(A\circ C\right)B^{T}$ coincide and we also used the fact that ${\frac{dG}{dx_{i}}}^{T}=\frac{dG}{dx_{i}}$ because of symmetry.

The matrix $R=\left(L⊙L^{-T}\right)$ is commonly known in the field of control theory as the relative gain array and admits many applications in the latter field.

Similarly,
(12)$$\frac{\partial \det(L)}{\partial z_{i}}=\frac{\det(L)}{\sigma_{d}^{2}}tr\left(L⊙L^{-T}\frac{dG}{dz_{i}}\right)$$

### 3.5. Proof of the Convergence of the Gradient Controller

We define the gradient controller as
(13)$$F_{\zeta }=H-\zeta \det\left(L\right)$$

To prove that our gradient controller (Equation (Equation 13)) converges to a critical point of $F_{\zeta }$, we must verify the following four properties:Must be differential;Must be locally Lipschitz;Must have a lower bound;Must be radially unbounded or the trajectories of the system must be bounded.

While this assures convergence to a critical point of $F_{\zeta }$, small perturbations to the system will cause the gradient controller to converge to a local minimum and not a local maximum or saddle point of the cost function.

*H* and det(L) verify properties 1, 2, 3, and 4.

We use also the sum of two Lipschitz functions asLipschitz.

Therefore, $F_{\zeta }$ verifies all the four properties [33].

## 4. Numerical Implementation and Experiments

To test the performance of our algorithm, we adopt the same environment parameters describing the concentration of CDOM specific to the depth of the Neponset River caused by the tide found in [25]. Each underwater environment is characterized by $\sigma_{s}$ and $\sigma_{d}$. Although those parameters were not explicitly given by the authors in their studies [25,34], we resort to a separability in the exponential function describing the covariance in order to extract them directly from Figure 4 in [34] via curve fitting.

For this first environment used in [34], we have $\sigma_{s}=2.074$ as covariance according to *X*, and $\sigma_{d}=0.917$ as covariance according to *Z*.

We use a grid size of length 8 km along the *X* direction and 3 m along the *Z* direction. Furthermore, we use a learning rate λ=0.1. Choosing an excessively large value of the learning parameter λ gives a wrong convergence and can make the system oscillate. However, choosing a too-small value λ makes the convergence sluggish.

Now, we report the experimental results for different number of sensors. Our second environment is characterized by $\sigma_{s}=1.977$ as covariance according to *X*, and $\sigma_{d}=1.198$ as covariance according to *Z*. We obtain similar results to environment 1. For the sake of brevity, we merely report the results for the second environment in Appendix A. Although we conducted a large set experiments for different sets of sensors and different parameters of the multiobjective function, we merely report a few representative results for the sake of brevity as the conclusions are similar for the different experiments. When it comes to the objective function, we report results for two representative cases: ω=0.8, which describes a case where the multiobjective function weights the covariance minimization term more, and ω=0.2, which describes a case where the multiobjective function favors the diversity maximization term more.

### 4.1. Case of 10 Sensors

In this scenario, we deploy 10 sensors initially at uniformly random positions and we run our scheme using ω=0.8 and ω=0.2. Note that according to the multiobjective function, ω=0.8 places more weight on the covariance, while ω=0.2 places more weight on the diversity.

Figure 3 shows the covariance after running our algorithm for $10^{4}$ iterations. We can clearly see that in the case of ω=0.2 we obtain the minimal covariance and fastest convergence rate.

Figure 4 shows that the corresponding diversity is largest for ω=0.2. We therefore conclude that by choosing ω=0.2, we obtain both lower covariance and higher diversity. In other terms, introducing the diversity term permits also to reduce the covariance as it seems that the diversity permits the optimization system to avoid some local minima.

The final positions are depicted in Figure 5. We visually observe that the positions with ω=0.2 give a total coverage of the sensors, while with ω=0.8, the sensors are positioned only in the middle and at the top of the network.

### 4.2. Case of 20 Sensors

Now, we describe the experiment for 20 sensors. We use the same values of ω=0.2 and ω=0.8 and show the graphs for covariance, diversity, and final positions.

The covariance is depicted in Figure 6 where the convergence speed seems faster for ω=0.2 compared to ω=0.8. The rate of diversity is depicted in Figure 7 for both values ω=0.2 and ω=0.8. We can see that ω=0.2 gives a higher value for the diversity. The final position of this case study is presented in Figure 8 and we can visually verify the adequate positioning of the sensors with ω=0.2, despite the increase in the number of sensors.

## 5. Further Discussion

As mentioned in Section 4, the convergence speed seems faster for ω=0.2 compared to ω=0.8 for both cases (10 and 20 sensors). By comparing the performances of two environments, we donate that for ω=0.2, the diversity seems to be faster than for ω=0.8 and the covariance seems to be minimal. In the performance comparison between two cases of study, 10 and 20 sensors and for ω=0.2 and ω=0.8, we notice that the covariance decreases rapidly only for the value ω=0.8, and for the diversity, the case ω=0.2 is more impacting than the case of ω=0.8. The positions of the sensors at the end show that for the value 0.2, we have more coverage of the study area than in the case of 0.8.

In Table 1, we give an overview of several papers published on quality of monitoring (QoM) in underwater sensors, each proposing a different approach and method for optimizing sensor placement and data collection in underwater environments.

These papers demonstrate that there are various approaches and methods for optimizing QoM in underwater sensor networks, and the performance of these methods can depend on factors such as the optimization algorithm, the network topology, and the environmental conditions. It is important to carefully consider these factors when designing and deploying underwater sensor networks and to evaluate the performance of different approaches using appropriate metrics and benchmarks.

## 6. Conclusions and Future Work

In this paper, a new optimization algorithm based on covariance and diversity is presented to optimize the QoM. Moreover, we presented the challenges associated with each of these blocks and how they were tackled by several relevant papers in the literature. This was performed in a systematic way, by focusing on the methods, conclusions, and higher level decisions of each paper. More specifically, we can conclude the following. (1) Input features should convey useful information about the propagation problem at hand, while also having small correlation between them. (2) Dimensionality reduction techniques can help identifying the dominant propagation-related input features by removing redundant ones. (3) Increasing the number of training data by presenting the ML model with more propagation scenarios improves its accuracy. As future work, we propose to investigate different aspects:The impact of varying the number of agents on the performance of the system: While our approach showed promising results in improving the performance of QoM, it is important to understand how the number of agents affects the overall system performance. Future work could focus on varying the number of agents and evaluating the resulting impact on the performance of the system.Agent selection: Future work could explore the development of a more efficient algorithm for agent selection that reduces computational costs while still achieving high-quality optimization results.Evaluating the impact of environmental factors on the performance of the system: The performance of underwater sensor networks is often affected by various environmental factors such as water temperature, salinity, and turbidity. Future work could investigate how these environmental factors affect the performance of the multiagent diversity-based gradient approach optimization and identify ways to mitigate their impact.Extending the optimization to other QoM metrics: The multiagent diversity-based gradient approach optimization has been mainly focused on optimizing the energy efficiency of underwater sensor networks. Future work could explore the extension of the optimization approach to other QoM metrics such as latency, throughput, and reliability. As future work, we could also try to jointly optimize the communication cost and quality of monitoring.Machine learning techniques such as reinforcement learning and deep learning have shown promising results in optimizing various aspects of underwater sensor networks. Future work could explore the integration of these techniques with our optimization approach to further improve the performance of the system.Three-axis models: To improve our study and move close to the real world, a three-axis model will be considered in future works.

## Figures and Tables

**Figure 1 sensors-23-03877-f001:**
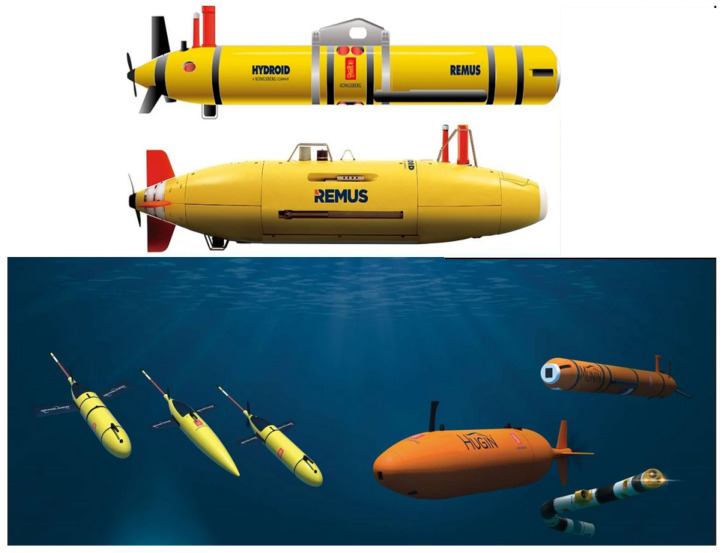
Illustrative examples of AUV systems.

**Figure 2 sensors-23-03877-f002:**
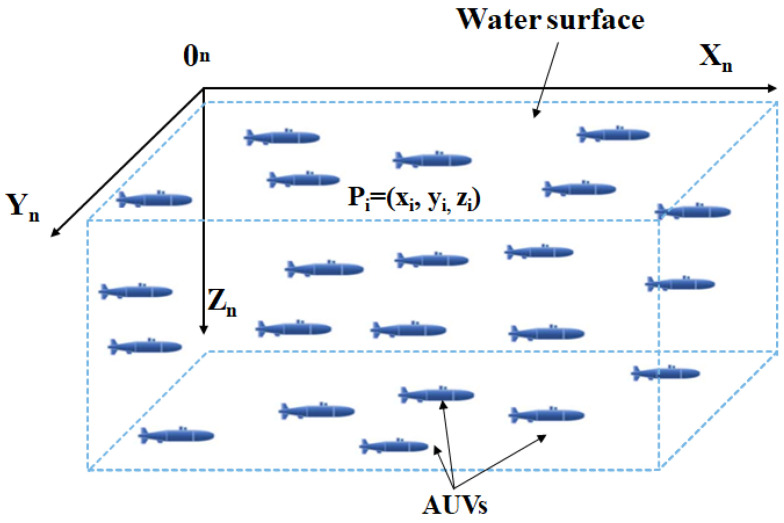
AUV system coordinates.

**Figure 3 sensors-23-03877-f003:**
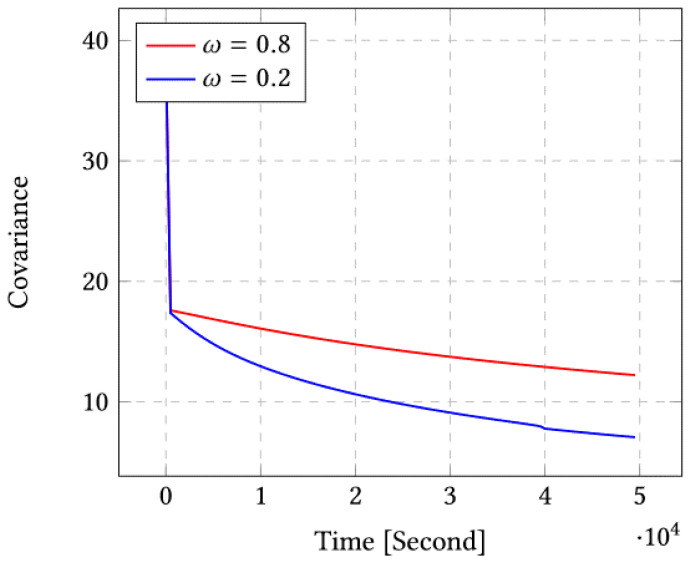
Covariance for ω=0.8 and ω=0.2 for the case of 10 sensors.

**Figure 4 sensors-23-03877-f004:**
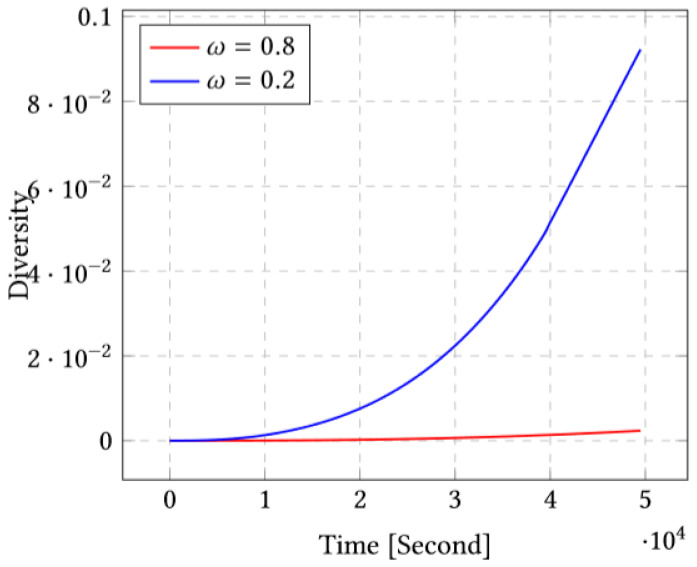
Diversity for ω=0.8 and ω=0.2 for the case of 10 sensors.

**Figure 5 sensors-23-03877-f005:**
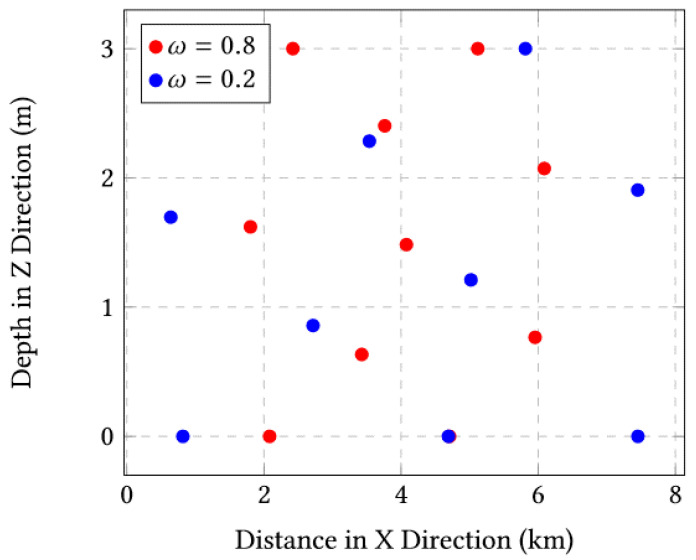
Final positions for ω=0.8 and ω=0.2 for the case of 10 sensors.

**Figure 6 sensors-23-03877-f006:**
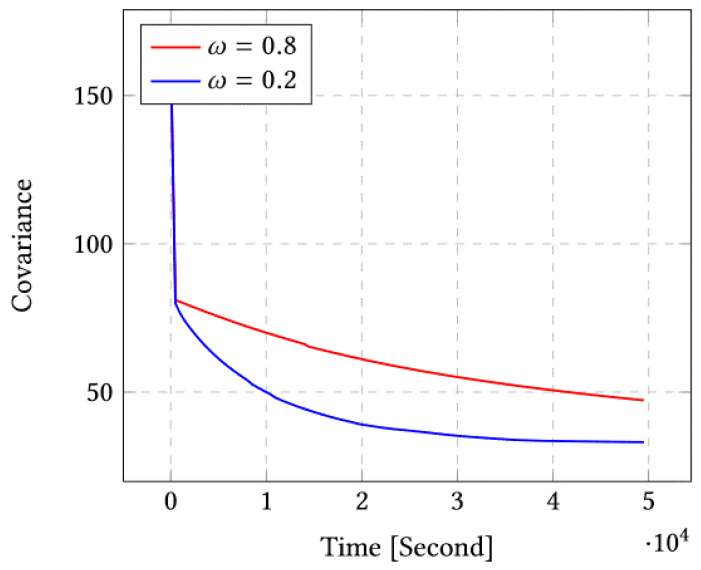
Covariance for ω=0.8 and ω=0.2 for the case of 20 sensors.

**Figure 7 sensors-23-03877-f007:**
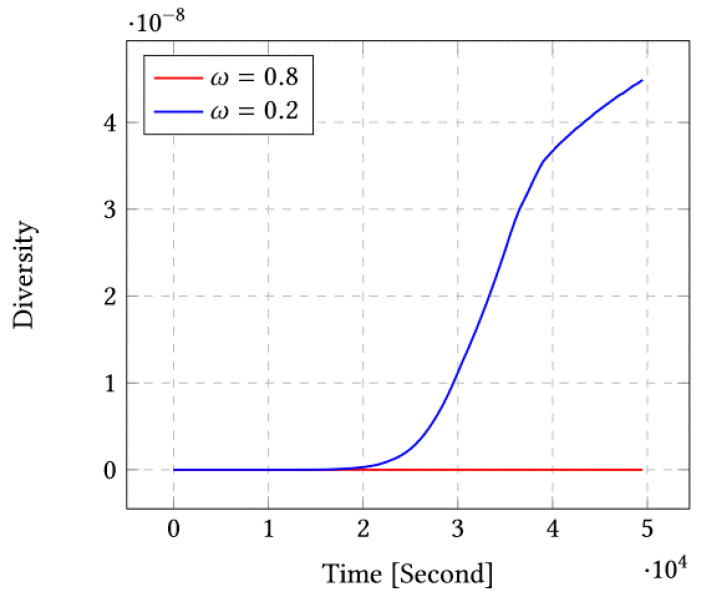
Diversity for ω=0.8 and ω=0.2 for the case of 20 sensors.

**Figure 8 sensors-23-03877-f008:**
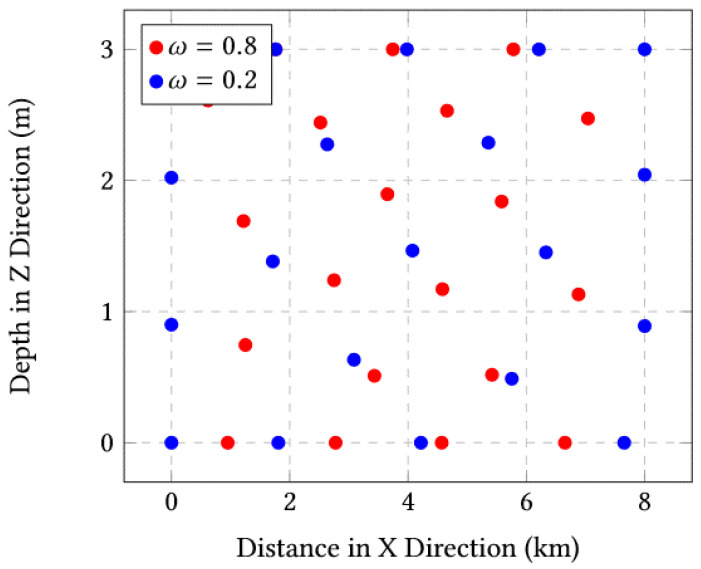
Final positions for ω=0.8 and ω=0.2 for the case of 20 sensors.

**Table 1 sensors-23-03877-t001:** Qualitative comparison of related works.

Reference	Used Approach	Optimization Algorithm	Objective Function
[25]	Multiagent approach	Gradient descent	Objective function based on the inverse of the sum of covariance between measured signals at sensors and points of interest
[38]	Algorithm based on Wolf Search	Wolf Search	Network coverage in terms of detection probability while accounting obstacle avoidance
[39]	Genetic algorithm-based	Elitist nonselective genetic algorithm (NSGAII)	Difference between measured signals and real signal generated by a known trajectory and a ferromagnetic object
[40]	DABVF, a distributed node deployment algorithm based on virtual forces	Virtual forces	Improve network coverage, reduce node energy consumption, balance node residual energy, and optimize node distribution
[41]	Multiagent target search method (MATSMI)	Multiagent deep deterministic policy gradient (MADDPG) method	Reward function per agent for finding target which decreases for longer discovery time
[42]	Range-based whale optimization algorithm	Whale optimization algorithm (WOA)	Best localization coverage, delivery ratio, delay, and energy
[43]	A nature inspired algorithm called underwater salp swarm algorithm (USSA)	Salp swarm algorithm	Maximize the number of localized nodes among nonlocalized ones
[44]	A hybrid optimization technique	Butterfly optimization and quaternion-based backtracking search optimization (QBSA)	Reducing the localization error based on the received signal strength indicator (RSSI), battery energy, and distance parameters
This Work	Multiagent system approach	Gradient descent	Overall covariance among measured signals by sensors and a diversity term

## Data Availability

Not applicable.

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
