# Peer review of "Quality of Monitoring Optimization in Underwater Sensor Networks through a Multiagent Diversity-Based Gradient Approach"

_sensors, 2023, doi:10.3390/s23083877_

Round 1

Reviewer 1 Report

First of all, I want to congratulate the authors for their efforts in this manuscript. It presents a novel idea, and it is interesting. The authors proposed an optimisation model for QoM in underwater networks in the manuscript. The paper is well-structured and exciting. Nonetheless, I have some concerns about the paper. First, the authors move the results of some of their simulations to the Anex to keep the paper for the sake of brevity. I am afraid I have to disagree with them. My other major concern is the reduced number of references in the paper. Following, I include the list of detected drawbacks:

-          In the introduction, more conceptualisation of some issues must be provided (at least 10 to 12 ref more):

o   More details on underwater acoustic communications (such as Sendra, S., Lloret, J., Jimenez, J. M., & Parra, L. (2015). Underwater acoustic modems. IEEE Sensors Journal, 16(11), 4063-4071. and Zia, M. Y. I., Poncela, J., & Otero, P. (2021). State-of-the-art underwater acoustic communication modems: Classifications, analyses and design challenges. Wireless Personal Communications, 116(2), 1325-1360)

o   More details of QoM and about water quality and ROVs

-          The authors must add the aim of the paper in a new paragraph at the end of the introduction. Consider using bullet points to highlight the novelty of the proposed solution. Part of this content is already in the related work.

-          Move the structure from the related work to the introduction

-          Add the content of the appendix to the paper and explain it in more detail.

-          Discussion must be extended, and compare their results with similar papers. Consider adding at least 10 to 15 references of similar and actual solutions.

-          Future work has to be presented in a separate paragraph and extended. I suggest including the 3 axis models in future work.

Reviewer 2 Report

I looked the article carefully “A Multi-Agent Diversity-based Gradient Approach for Quality
of Monitoring Optimization in Underwater Sensor Networks”.  The topic is useful to research community and interesting. My concerns are:

·         Title of manuscript should be revised as it is unclear according to work.

·         Abstract and conclusion are to general, these should be specified with some focused information.

·         Appendix A. derivation of the gradient of diversity should be add in methodology or removed from manuscript  

·         Extensive discussion required before conclusion.

·         At the end needs to add comparison of results with literature. How this work is different? It would be good if authors add bench mark table before conclusion that will show comparison of results.

Round 2

Reviewer 1 Report

The authors have to provide an adequate response to the comments I included. Indicating which changes have been made. In the manuscript, I cannot see the modifications since they are not highlighted. 

After checking the manuscript, I found that some of my comments are not correctly addressed. 

Round 3

Reviewer 1 Report

The authors have addressed the comments.